# 5-Aryl-2-(3,5-dialkyl-4-hydroxyphenyl)-4,4-dimethyl-4*H*-imidazole 3-Oxides and Their Redox Species: How Antioxidant Activity of 1-Hydroxy-2,5-dihydro-1*H*-imidazoles Correlates with the Stability of Hybrid Phenoxyl–Nitroxides

**DOI:** 10.3390/molecules25143118

**Published:** 2020-07-08

**Authors:** Svetlana A. Amitina, Elena V. Zaytseva, Natalya A. Dmitrieva, Alyona V. Lomanovich, Natalya V. Kandalintseva, Yury A. Ten, Ilya A. Artamonov, Alexander F. Markov, Dmitrii G. Mazhukin

**Affiliations:** 1Novosibirsk Institute of Organic Chemistry, Siberian Branch of Russian Academy of Sciences (SB RAS), Academician Lavrentiev Ave. 9, 630090 Novosibirsk, Russia; amitina.sa@gmail.com (S.A.A.); elena@nioch.nsc.ru (E.V.Z.); loman@nioch.nsc.ru (A.V.L.); ten@nioch.nsc.ru (Y.A.T.); artamonov198888@mail.ru (I.A.A.); 2Department of Chemistry, Institute of Chemistry of Antioxidants, Novosibirsk State Pedagogical University, Vilyuyskaya Str. 28, 6301026 Novosibirsk, Russia; n_gaas@mail.ru (N.A.D.); aquaphenol@mail.ru (N.V.K.); chemistry@ngs.ru (A.F.M.)

**Keywords:** cyclic hydroxylamines, 4*H*-imidazole 3-oxides, sterically hindered phenols, antioxidants, hybrid phenoxyl–nitroxides, electron paramagnetic resonance

## Abstract

Cyclic nitrones of the imidazole series, containing a sterically hindered phenol group, are promising objects for studying antioxidant activity; on the other hand, they can form persistent hybrid phenoxyl–nitroxyl radicals (HPNs) upon oxidation. Here, a series of 5-aryl-4,4-dimethyl-4*H*-imidazole 3-oxides was obtained by condensation of aromatic 2-hydroxylaminoketones with 4-formyl-2,6-dialkylphenols followed by oxidation of the initially formed *N*-hydroxy derivatives. It was shown that the antioxidant activity of both 1-hydroxy-2,5-dihydroimidazoles and 4*H*-imidazole 3-oxides increases with a decrease in steric volume of the alkyl substituent in the phenol group, while the stability of the corresponding HPNs generated from 4*H*-imidazole 3-oxides reveals the opposite tendency.

## 1. Introduction

Nitrones constitute a rapidly emerging class of functional substances whose high reactivity not only serves as a starting point for the synthesis of biologically important nitrogen compounds [1,2] but also perfectly correlates with the manifestation of a wide range of bioactivities by them [3,4,5]. Nitrones are unique free-radical scavengers acting as spin traps of very cytotoxic reactive oxygen species (ROS) accumulating in tissues at high concentrations under oxidative stress.

High antioxidant activity of nitrones bearing aliphatic, aromatic, heterocyclic, or heteroatom substituents allows the consideration of them as effective therapeutic agents against oxidative stress that occurs under critical pathological conditions in the human body, e.g., myocardial infarction or stroke [6]. One of the first nitrones to be used as a neuroprotector reducing ischemic hippocampal damage in animals was *N*-*tert*-butyl-α-phenylnitrone (PBN) **1a** [7]. Furthermore, a series of **1a** derivatives has been synthesized and applied as (i) hydrophilic spin traps for different types of organic radicals (e.g., 4-POBN **1b** for OH^•^, CO_2_^-^^•^) [8] and (ii) spin traps for the hydroxymethyl radical and promising agents for neuroprotection yielding good results in an in vitro model of cellular injury of cortical neurons (*N*-aryl and *C*-alkyl modified nitrones **3a**,**b**) [9]. Water-soluble disulfonate PBN derivative **1c** (NXY-059) [10] has reached AstraZeneca Phase IIb/III clinical trials for the treatment of acute stroke, and although it eventually failed at the efficacy assessment when compared with a placebo, at the same time, this compound turned out to be a promising novel anticancer agent (OKN-007) [11]. Lipophilic bis-nitrone of the azulene series **2** (STAZN), possessing an improved level of blood-brain barrier penetration, has shown impressive results in terms of both outstanding antioxidant activity [12] and prospects for its application as a lead compound for stroke treatment [13] (Scheme 1).

The chemical and pharmacological properties of nitrones depend mainly on the bioavailability as well as the nature and position of the substituents in the nitrone group. During the last decade, Marco-Contelles at al. have synthesized and tested a large series of acyclic nitrones that hold promise for the treatment of neurodegenerative diseases: (a) PBN analogs containing isovanillin moieties (RP-6) **1d** [14], (b) 3- and 4-nitronylquinolines **5** (RP-19 and QN-23) [15,16] and **6** (QN-6) [17], and (c) heteroatomic *C*-phosphoryl derivative **3c** [18] (Scheme 1). Moreover, very recently, a new cholesterol derivative containing exocyclic nitrone function **4** (ISQ-201) was obtained by the same group of researchers [19]. Detailed study of two spatial isomers of this lipophilic molecule has revealed that only the (*E*)-isomer significantly decreases ischemia-induced neuronal death and apoptosis, in a dose-dependent manner. This compound manifested its therapeutic effect when administered before 6 h after postischemic reperfusion onset: effects that persisted for 3 months after the ischemic episode [20]. In addition, other heterocyclic nitrones of pyrazine **7** (TBN) [21] and triazole **8** [22] series have shown noticeable anti-inflammatory and anticarcinogenic properties.

Among cyclic nitrones, derivatives of pyrroline-*N*-oxide **9** are best known as spin traps of free radicals, such as 5,5-dimethyl-1-pyrroline *N*-oxide (DMPO) [23] (to date, over 3000 papers on DMPO have been published) and its functional derivatives, ethoxycarbonyl (EMPO) [24], the more lipophilic *t*-butoxycarbonyl (BMPO) [25], and 5-diethoxyphosphoryl-5-methyl-1-pyrroline 1-oxide (DEPMPO) [26], which have contributed substantially to the understanding of free-radical-mediated processes in biochemical systems. Some cyclic nitrones can trap different short-lived radical species, forming more stable spin adducts than PBN or DMPO. These include compounds of the imidazole series **10** [27,28] and **14** [29], isoindoline **12** [30], and isoquinoline **13** [31] derivatives. Screening of recently synthesized 3*H*-indol-3-one 1-oxides revealed that 2-alkylisatogens **11** are novel ROS scavengers capable of inhibiting cellular necroptosis [32], whereas 2-aryl derivatives possess very promising antiplasmoidal activity [33] (Scheme 2).

Although a wide range of nitrones has been obtained, and their biological activity in various models has been proven, the question of the correlation between their ability to capture short-lived radicals and their neuroprotective activity is still open. As recently noted, “the mode of action of nitrone has been subject to debate over the past three decades and the exact mechanism of neuroprotection is not fully known. There is strong evidence that nitrones may affect genomic regulation and stimulate anti-inflammatory reactions as well as antioxidant properties and action on important membrane enzymes. However, so far there is no clear link between the chemical structure, the chemical reactivity and the potency of nitrone in preventing cell death.” [9]. Therefore, the search for new biologically active functional nitrones is still a relevant and important task.

As part of our ongoing research on the properties and applications of π-conjugated planar stable hybrid phenoxyl–nitroxides (HPNs) based on the 4*H*-imidazole 3-oxide frame [34,35], we turned our attention to the possible application of diamagnetic HPN precursors: 2,4-diarylimidazole derivatives bearing either hydroxylamine or nitrone functions—along with the sterically hindered phenolic group—as promising antioxidants containing two antiradical moieties. Although only a limited number of compounds possessing a structure of similar type (compounds **15–19)** is described in the literature, they have proven to be “dual” traps of free radicals (**15** [36,37,38] and **16** [39])—in addition to having remarkable properties—and as unique agents with a wide spectrum of biological activities, in particular, antioxidant, anti-inflammatory, and neuroprotective properties (**15**, **17**, **18**) (Scheme 3) [40,41,42,43,44]. Hybrid molecules **19** (R = *t*-Bu, cyclopropyl), combining a vitamin E moiety and a spin trap part, as it has turned out, are comparable to Trolox in scavenging free radicals and outperform nitrone-type reference compounds, for example, PBN or NXY-059 [45].

In this regard, one of the purposes of the present study was to determine a possible mutual influence of the nitrone (or cyclic hydroxylamine) group and the sterically hindered phenolic moiety introduced into the frame of azoles **20** and **21** on their antioxidant (antiradical) activity. Another objective was to study the effect of steric and electronic factors of substituents on the stability of HPNs formed upon the oxidation of 4*H*-imidazole 3-oxide derivatives **21**. The latter topic is important in the sense that conjugated planar hybrid organic radicals are of interest as promising building blocks for the design of functional magnetic materials [46,47,48], and the effectiveness of the intermolecular spin–spin exchange interaction depends on the packing in the crystal (this packing is determined by the geometry of the substituents and functional groups).

In this work, we report: (i)the synthesis of 2-(3,5-dialkyl-4-hydroxyphenyl)-4,4-dimethyl-substituted imidazole derivatives **20** and **21**(ii)assessment of their antiradical activity toward the model reaction of cumene oxidation, and(iii)4*H*-imidazole 3-oxide **21–**based preparation of new persistent HPNs **22** and their characterization by EPR in solution.

## 2. Results

### 2.1. Preparation of Key Compounds

#### 2.1.1. Synthesis of 4-formylphenols **23** and Their Condensation with *p*-X-Ar-substituted 2-hydroxylamino Ketones **25**

4-Formyl-2,6-dialkylphenols **23a–e** required for the further condensation with 2-hydroxylamino ketones were obtained with high yields via the Duff reaction of α,α′-dialkyl–substituted phenols **24a–e** with an excess of hexamethylenetetramine, during heating in glacial acetic acid (Scheme 4).

To synthesize the main synthetic blocks (2-hydroxylamino ketones **25)**, α-bromo derivatives of *p*-aryl–substituted isobutyrophenones **26** were reacted with nitrogen nucleophiles in two ways outlined in Scheme 5. Following the first method, bromoketone **26d** containing a deactivated phenolic hydroxyl group was introduced into the reaction with an excess of a sodium salt of *Z*-benzaldoxime (cf. [49]) and the obtained nitrone **27** was subsequently treated with 1 equiv of NH_2_OH⋅HCl. After extraction of the reaction mixture and separation of waste benzaldoxime, the aqueous solution was carefully neutralized by ammonia to quantitatively precipitate hydroxylamino ketone **25**. According to the second method, when reactive bromoketones **26a–c** were introduced into the reaction with a large excess of hydroxylamine, intermediate 2-hydroxylamino oximes **28a–c** were obtained with a high yield. Boiling of the latter in concentrated hydrochloric acid led to the desired 2-hydroxylamino ketone **25a–c**, as their corresponding hydrochlorides (Scheme 5).

Condensation reactions of resultant hydroxylamino ketones **25a–c** (as hydrochlorides) or as free bases (**25d**) with 4-formylphenols **23a–e** were carried out in the presence of a large excess (6–10 equiv) of ammonium acetate to minimize the formation of the side products of the reaction, ketonitrones **29**, and therefore to increase the yield of the desired 1-hydroxy-2,5-dihydroimidazoles **20a–s** (Scheme 5, Table 1). Although this reaction takes place at ambient temperature and usually finishes in 6–12 h, we noticed that overexposure of the mixture in an open flask leads to its darkening and decreases the yields of imidazolines **20a–s** owing to oxidative processes. Precipitated cyclic hydroxylamines **20** were pure enough to use them for the next synthetic step; the samples designed to investigate their antioxidant properties were purified by crystallization from an appropriate solvent (Appendix A).

#### 2.1.2. Oxidation of 2,5-dihydroimidazoles **20** to 4*H*-imidazole 3-Oxides **21**

4*H*-Imidazole 3-oxides **21a–s** were prepared with a high yield via oxidation of cyclic hydroxylamines **20a–s** by air oxygen using a mild homogeneous catalytic system: a copper(II) ammine complex in aqueous methanol (Scheme 5, Table 1). Attempts to apply various oxidants (heterogeneous MnO_2_, PbO_2_, or homogeneous aqueous solutions of sodium periodate or alkaline potassium hexacyanoferrate) to this reaction led to a lower product yield and significant resinification of the reaction mixture owing to side oxidation of the phenolic hydroxyl group. 4*H*-Imidazole 3-oxide derivatives **21a–s** are bright yellow or orange high-melting-point crystalline compounds, poorly soluble in polar and aprotic solvents and moderately soluble in halogenated hydrocarbons.

### 2.2. Antiradical Activity of the 1-Hydroxy-2,5-Dihydroimidazole **20** and 2,5-Diaryl-4H-Imidazole 3-Oxide **21** Derivatives

Comparative analysis of antiradical activity (ARA) of the 27 synthesized imidazole derivatives **20–21** was carried out in the model system of azobisisobutyronitrile (AIBN)-initiated cumene oxidation at 60 °C. The oxidation of aliphatic and aromatic hydrocarbon derivatives, polymers, and lipids by molecular oxygen is a radical chain process proceeding according to the hydroperoxide mechanism, which can be described by means of reactions (0)–(6) (Appendix A). Interaction of the phenolic antioxidants with peroxide radicals of an oxidizable substrate (Appendix A Equation (1)) is the principal step determining the ability of phenolic compounds (ArO-H) to inhibit the chain oxidation process. Unlike the peroxide radical RO_2_^•^, phenoxyl radical ArO^•^ formed in reaction (7) is inactive in the chain extension process; therefore, the presence of ArO-H significantly decelerates the substrate oxidation. Thus, the rate constant (*k_7_*) of the reaction between the antioxidant (AO) and peroxide radicals is one of the main characteristics of the effectiveness of an inhibitor. Therefore, *k*_7_ as well as stoichiometric inhibition coefficients *f*, numerically equal to the average number of oxidation chains terminated per phenoxyl group of the inhibitor, were chosen as the quantitative ARA characteristics.

According to the obtained data (Table 2), all the investigated compounds had a pronounced inhibitory activity against the oxidation of cumene. Moreover, different groups of compounds showed different inhibition coefficient values *f* and the rate constants *k_7_* as well.

Thus, for 2,5-dihydroimidazoles **20**, inhibition coefficient *f* was 2.8 to 4.8, but 4*H*-imidazole 3-oxides **21** were characterized by lower values of *f* (1.9 to 2.5) under the same experimental conditions.

It is known that an *f*-value experimentally determined under conditions of initiated oxidation is close to 2 for most of 2,4,6-trialkylphenols [50], which corresponds to quantitative transformation of phenols (ArO-H) to phenoxyls (ArO^•^) via Equation (1) (Appendix A), followed by conversion of the latter into the molecular products by Equation (2) (Appendix A):ArO-H + RO_2_^•^ → ROOH + ArO(1)
ArO^•^ + RO_2_^•^ → molecular products(2)

It is noteworthy that a clear dependence of the experimentally determined *f*-value on the nature of the *ortho*-substituents in the hydroxyaryl part of 2,5-dihydroimidazoles **20** was observed. Indeed, compounds **20e**,**j**,**o** with di-*tert*-butyl *ortho*-substituents near phenolic hydroxyl, were characterized by lower *f*-values (2.8 to 3.2) than their analogs bearing methyl, isopropyl, and cyclohexyl as *ortho*-substituents, featuring *f*-values of 3.9 to 4.9 (Table 2).

Because the *f*-values of compounds **20a,f,k** with the least bulky methyl *ortho*-substituents did not exceed those of isopropyl and cyclohexyl-substituted analogs, it can be assumed that for the manifestation of high *f*-values in the series of 2,5-dihydroimidazoles, the presence of *ortho*-alkyl substituents with benzylic hydrogen atoms is crucial. In other words, in the case of *tert*-butyl-substituted **20e,j,o** the oxidation of the hydroxyaryl moiety was stopped at the stage of phenoxyl radical formation, whereas for methyl-, isopropyl-, and cyclohexyl-substituted derivatives, the interaction with peroxide radicals was more profound, leading to the formation of substituted *ortho*-methylenequinones. Moreover, 1-hydroxy-2,5-dihydroimidazoles **20** can break the oxidation chains via a reaction involving the *N*-hydroxy groups. The formed nitroxyl apparently does not have an inhibitory activity. This assumption is supported by the established fact that 4-hydroxy-2,2,6,6-tetramethylpiperidine 1-oxyl (TEMPOL) does not inhibit initiated oxidation of cumene [51]. Finally, one more antiradical center in **20** is obviously the hydrogen atom at the C-2 position of the imidazole ring. The easiness of the cleavage of the corresponding C-H bond is due to the stability of the forming *C*-centered radical, containing a prolonged conjugated system covering both aryl substituents of the imidazole ring and providing effective delocalization of the spin density of an unpaired electron. Recombination of the peroxide radical with the *C*-centered radical breaks another oxidation chain and leads to the final product. Accordingly, in the reaction with peroxide radicals one 2,5-dihydroimidazole molecule **20** can interrupt at most five oxidation chains, turning presumably into 3-imidazoline 1-oxyl **30** (Scheme 6).

The experimentally obtained *f*-values for 2,5-dihydroimidazoles **20g**,**l**,**n** amounted to 4.7–4.8, respectively, and were in good agreement with the above-mentioned reasoning. The most likely final products of the oxidation of compounds **20e**,**j**,**o** are corresponding stable HPNs **22**, bearing a quinone methide–conjugated moiety. Conversion from **20e** to **22e** corresponds to breakage of 3 oxidation chains, in good agreement with the obtained *f*-values for **20e**, **20j**, and **20o** (2.8 ± 0.3 to 3.2 ± 0.3).

The use of 4*H*-imidazole 3-oxide derivatives **21** as AOs in comparison with 2,5-dihydroimidazoles **20** under the same experimental conditions caused less prolonged inhibition of cumene oxidation. Indeed, the *f*-values of **21** were within the range 1.9 to 2.5, wherein, similarly to compounds **20**, derivatives **21e**, **21j**, and **21o** with a di-*tert*-butyl-substituted hydroxyaryl part were characterized by lower *f*-values equal to 1.9–2.0. At the same time, the methyl-, isopropyl-, and cyclohexyl-substituted analogs had *f*-values of 2.05 to 2.5.

Molecules of 4*H*-imidazole 3-oxide **21** contain two antiradical centers: a hydroxyaryl part and a nitrone group, respectively. It might be suggested that the hydroxyaryl group in **21** bearing *ortho*-substituents with benzylic hydrogen atoms can undergo transformations similar to those for 2,5-dihydroimidazoles **20**, interrupting two oxidation chains and, consistently turning into the corresponding phenoxyl radicals and then into *ortho*-methylenequinones. On the other hand, nitrones are also able to break oxidation chains via the addition of active radicals to a double bond with the formation of stable nitroxide radicals [52], which in the case of **21** leads to analogous paramagnetic product **30**. Therefore, it is expected that 4*H*-imidazole 3-oxides **21** bearing partially shielded hydroxyaryl substituents can interrupt three oxidation chains at most. This notion is in good agreement with the experimentally obtained data for this series of compounds **21k–n**: the *f*-coefficient reached 2.5, which is close to 3.0.

Compounds **20** and **21** contain several reaction centers able to interact with cumene peroxide radicals, but only one of them—the 3,5-dialkyl-4-hydroxyphenyl moiety—is common for both types of molecules. Because significant differences in *k_7_*-values, characterizing the corresponding pairs of compounds, 2,5-dihydroimidazole **20** and 4*H*-imidazole 3-oxide **21** series, were not observed, we assumed that the experimentally obtained values of *k_7_* (Table 2) refer to 3,5-dialkyl-4-hydroxyphenyl parts of these compounds.

It is known [50] that the reactivity of *ortho*-dialkyl–substituted phenols toward active radicals depends, on the one hand, upon the steric hindrances created by *ortho*-substituents for the ArO-H bond attack (that is, upon the effective volume of the *ortho*-substituents), and, on the other hand, upon the energy of the ArO-H bond, which decreases as the electron-donating ability of *ortho*-substituents increases. These factors are independent, thus often making quantification of the alkyl substituents’ effect on the *k_7_* value difficult. It has been shown previously that *ortho*-di-*tert*-butyl-substituted phenols are inferior in the magnitude of *k_7_* to their analogs containing *n*-alkyl and *sec*-alkyl *ortho*-substituents. According to the literature data, *k_7_* values of 2,4,6-trialkyl–substituted phenols are 2.4 × 10^4^ to 1.6 × 10^5^ M^−1^**•**s^−1^ under the conditions of initiated cumene oxidation at 60 °C. Herewith, 2,6-di-*tert*-butylphenols were characterized by the lowest values of *k_7_*, which were 5–7-fold less than those for their analogs with dimethyl and dicyclohexyl *ortho*-substitution [53].

The measured *k_7_* values for imidazole derivatives **20** and **21** under study also changed depending on the nature of the alkyl *ortho*-substituents; however, the differences in the determined values were much less pronounced. Specifically, for 2,5-dihydroimidazole series **20**, the compounds with *tert*-butyl substituents were characterized by *k_7_* values equal to (4.0–4.7) × 10^4^ M^−1^**•**s^−1^, and the highest *k_7_* values were detected for dimethyl-substituted derivatives **20a**, **20f**, and **20k**: 4.9 × 10^4^ to 6.4 × 10^4^ M^−1^**•**s^−1^. A similar trend was also observed for 4*H*-imidazoles **21** (Table 2). In this way, the presence of the 3,5-dialkyl-4-hydroxyphenyl moiety at the C-2 atom of the imidazole ring in **20** and **21** causes leveling of differences in *k_7_* values characterizing the ARA of ArO-H with *ortho*-substituents of different nature. The presence or absence of the halogen atom at the 4(5)-position of the heterocycle did not affect the experimentally determined values of the *k_7_* constant, most likely owing to the significant remoteness of the functional group from the active center of the phenolic part.

### 2.3. Generation and EPR Study of New Hybrid Phenoxyl–Nitroxides **22**

Recently, we published a detailed study on intra- and intermolecular magnetic properties of HPNs bearing an aliphatic, aromatic, and heteroaromatic substituent at the C-4(5) carbon atom of the heterocyclic ring to show their usefulness as new building blocks for molecular magnetic materials [35]. To continue the detailed analysis of magnetic properties of this new class of organic radicals for their further potential application in magnetochemistry, we investigated in the present study how structural features of the molecule, in particular, the nature of the substituents adjacent to the phenolic group, can affect the stability and electronic properties of persistent HPNs **22** (Figure 1) by means of X-band continuous-wave (CW) EPR spectroscopy and density-functional theory (DFT) calculations.

HPNs **22a–o** were generated via oxidation of 4*H*-imidazole 3-oxide derivatives **21a–o** by lead dioxide in diluted toluene solutions (see the Experimental section for details). The corresponding EPR spectra were recorded after the oxidant filtration and subsequent deoxygenation of the obtained radical solutions. The spectra of **22a–d** and **22e**,**j**,**o** are presented in Figure 2 and Figure 3, respectively; all the other spectra can be found in Appendix A. The observed spectra represent complex patterns at g_iso_ = 2.0049 to 2.0063, which can be well reproduced taking into account hyperfine splitting *(hfs)* constants of two nonequivalent nitrogen nuclei (N^1^ and N^3^), two slightly different phenoxyl protons (H^2^ and H^6^), and the protons of alkyl substituents at α,α′-carbon atoms of the phenoxyl group. The simulation parameters are listed in Table 3. The observed *hfs* constants varied within the following limits: A_N1_ = 0.343–0.550 mT, A_N3_ = 0.042–0.064 mT, A_H2, H6_ = 0.066–0.312 mT, and A_H-R1R2_ = 0.150–0.767 mT, in line with the quantum chemical calculations presented below.

It is worth noting that the structure of alkyl substituents at the 3rd and 5th positions of the phenoxyl moiety significantly influences the spin density distribution within the phenoxyl-nitroxide part (blue in Figure 1). Consequently, the main part of the spin density in hybrid radicals **22a**, **f**, **k** with primary alkyl groups (R^1,2^ = CH_3_) is localized on the methyl moieties. Indeed, the observed *hfs* constants of the methyl protons are double the *hfs* constant of the nitrogen nuclei of nitroxide groups. When the substituents at C-3,5 of the phenoxyl ring are replaced with tertiary alkyl groups (compounds **22b–d**, **g–i**, **l–n**), the *hfs* constants of the methine hydrogen atom of the cyclohexyl or isopropyl substituent decrease 3.5-fold in comparison with those for methyl-substituted analogs and, therefore, they become 2- to 3-fold less than the *hfs* constant of the nitrogen nucleus of the nitroxide part and 1.5-fold less than *hfs* constants of phenoxyl protons H^2,6^. This is due to the presence of several conformational isomers of these radicals with different spin density distributions, which exist in equilibrium in their diluted solutions, and therefore the observed spectra are a superposition of the spectra of these conformers.

Due to the absence of protons at α,α′-carbon atoms, radicals bearing *tert*-butyl substituents R^1^ and R^2^ (compounds **22e**,**j**,**o**) have *hfs* constants of only two nonequivalent nitrogen nuclei of the heterocycle and two nonequivalent protons H^2,6^ (Figure 3). In addition, it was possible to estimate a small *hfs* constant of the fluorine nucleus for radical **22j**. Moreover, slight differences in the chemical structures of **22e**,**j**,**o** led to changes of their EPR spectra and thus allowed to analyze how the R^3^ substituent (H, F, Br) influences the spin density distribution within the side part of the molecules. Nevertheless, a comparison of *hfs* constants of these radicals showed that in general, the spin density distribution map changed only insignificantly. Thus, *hfs* constants of **22e**, **j**, **o** vary within the following limits: A_N1_ = 0.548–0.552 mT, A_N3_ = 0.061–0.063 mT, A_H2_ = 0.163–0.166 mT, and A_H6_ = 0.152–0.158 mT. The obtained data are well consistent with the theoretical calculations described below.

To additionally confirm our interpretation of the EPR spectra of HPNs **22a–e**,**j**,**o**, we calculated their *hfs* constants by the DFT/UB3LYP/6-31G(d) method. The conductor-like polarizable continuum model (CPCM) (solvent: toluene) was employed to take into account a solvent effect. The calculation results are listed in Table 4 and presented in Appendix A. In general, the calculation data qualitatively correlated with the data obtained by the experimental spectra simulations. As shown in Appendix A, there are *hfs* constants of only two protons of methyl groups at α,α′-carbon atoms of the phenoxyl part, which are located outside the plane of the aromatic ring. It is clear that in solution, due to the free rotation of this group, an average value of these constants will be observed. This value, calculated as the arithmetic mean of the constants of three protons, is given in Table 4. Similar patterns were observed for HPNs **22b**,**c**,**d** containing cyclohexyl and isopropyl groups (Appendix A) and existing in solutions as equilibria of conformational isomers (rotamers). For those conformers, where the methine proton in the cyclohexane or isopropyl moiety is located in the plane of the phenoxyl ring, its *hfs* constant is significantly less than that when the proton is located out of the plane. The calculation method used in this work does not allow estimation of the molar ratio of the conformers in the mixture; therefore, the *hfs* constants listed in Table 4 were calculated assuming that the conformers exist in the mixture in the equimolar ratio.

As mentioned above, EPR spectra of **22e**,**j**,**o** are slightly different. Appropriate quantum chemical calculations were performed to investigate how the R^3^ substituent (H, F, Br) influences the spin density distribution in the side part of the molecule of HPN. Mulliken atomic spin populations for **22e**, **j**, **o** are presented in Figure 4. It was confirmed by the calculations that the origin of the substituent does not significantly influence the spin density distribution, in agreement with the data obtained by the analysis of their EPR spectra. The calculated spin densities on the aromatic ring vary within the range −0.023 to −0.025 for *ortho*- and *para*-carbons and 0.012–0.013 for *meta*-carbons, respectively; however, the sign of the *hfs* constant changes when a hydrogen atom is replaced by a halogen atom. The latter can be important for studies on the magnetic properties of these compounds in crystal, because at a certain mutual orientation of the molecules, the sign of the corresponding intermolecular magnetic interaction may change.

Thermodynamic stability of organic radicals (from persistence in solution to high robustness in a crystal state) is an important parameter for their any applications in the chemistry of magnetic and electroactive materials; therefore, we conducted its comparative assessment for a series of HPNs **22**. EPR spectra of the diluted toluene solutions of phenoxyl–nitroxides **22a–o** were recorded just after the formation of the corresponding radical and after keeping the solution for 1 h at 295 K in an Ar atmosphere. It was revealed that compounds **22a–d**, **f–i**, **k–n** containing similar primary or tertiary substituents at C-3(5) atoms on the phenoxyl ring are persistent and exist in solution for several hours. The least stable were radicals **22b**, **g**, **l** bearing asymmetric substituents: methyl and cyclohexyl groups. Indeed, after incubation of their solution for 1 h, only half of the initial amount of the radical remained. Surprisingly, phenoxyl–nitroxides **22a**, **f**, **k** with two methyl groups in the phenoxyl moiety were comparable in stability with **22c**, **d**, **h**, **i**, **m**, **n** containing isopropyl and cyclohexyl substituents at the same positions. Nonetheless, measurement of the decomposition kinetics for these compounds was challenging, because the EPR spectra of these compounds represent a superposition of two spectra: the spectrum of HPN (**22a**, **f**, **k**) and the spectrum of the paramagnetic product of its decompositions: a nitroxide radical of unknown structure, whose contribution is quite substantial (Figure 5). The ratio of intensities of these two spectra is time-dependent because the unknown radical gradually turns into diamagnetic products. In the case of HPNs **22c**, **d**, **h**, **i**, **m**, **n**, the contribution of the paramagnetic impurity is small and can be ignored in the analysis of their decomposition kinetics.

Accordingly, the spectra of diluted toluene solutions of **22c**, **d** were recorded every 10 min during 4 h. Double integrals of these spectra were computed to estimate concentrations of the paramagnetic compounds in solutions. The obtained kinetic curves are presented in Figure 6. Readers can see that the radicals decompose via first-order kinetics at approximately the same rate: the rate constants were found to be 1.2 × 10^−4^ s^−1^. Radicals **22e**, **j**, **k** with quaternary substituents in the phenoxyl moiety are quite stable compounds. They can be isolated as solids and stored for a long time under ambient conditions without changes.

Attempts to oxidize 4*H*-imidazole 3-oxides **21p–s** containing a *para*-hydroxy group in the aryl substituent at the C-5 atom (R^3^ = OH) did not lead to the formation of corresponding persistent HPNs **22p–s**. The reason is possibly the fast reduction of the formed paramagnetic center as a result of its intermolecular interaction with a labile phenolic group.

## 3. Materials and Methods

### 3.1. General Information

Fourier transform infrared spectra (FT-IR) were acquired in KBr pellets on a Bruker Vector-22. The UV-Vis spectra were obtained for EtOH solutions of 4*H*-imidazole 3-oxides **21** using a Hewlett-Packard HP 8453 spectrophotometer. ^1^H nuclear magnetic resonance (NMR) and ^13^C NMR spectra were recorded on Bruker AV-300, AV-400, and DRX-500 spectrometers at 300/400/500 and 75/100/125 MHz, respectively, for 3–10% solutions of compounds in CDCl_3_ and DMSO-*d*_6_; the positions of signals were determined relative to residual proton signals [DMSO-*d*_6_ (2.50 ppm), CDCl_3_ (7.24 ppm) for ^1^H spectra] or carbon signals [DMSO-*d*_6_ (39.4 ppm), CDCl_3_ (76.9 ppm), for ^13^C spectra] of the deuterium solvent. The assignment of signals of carbon atoms in the ^13^C NMR spectra of compounds **20**, **21** was made on the basis of a previous spectral work on the spectra of cyclic nitrones of the 4*H*-imidazole series [54]. Elemental analyses were performed on an automatic CHNS analyzer Euro EA 3000. The melting points were determined by means of an FP 81 HT instrument, Mettler Toledo. Column chromatography and thin-layer chromatography (TLC) were performed using Acros silica gel 60A (0.035–0.070 mm) and Sorbfil PTLC-AF-UV 254 (Russia), respectively, eluents: CHCl_3_, CHCl_3_-MeOH.

ARA of compounds **20**, **21** was measured in the model system of initiated AIBN cumene oxidation at 60 °C. The intensity of oxidative processes was monitored by means of the rate of oxygen uptake, the volume of which was measured on a Warburg-type apparatus by the method described earlier by Tsepalov [55]. Plotting and mathematical processing of kinetic curves were carried out in Origin 6.0 software.

The working concentrations of the components in a sample (60 °C) were as follows; RH: 6.9 M, AIBN: 5.3 × 10^−3^ M, AO: 5 × 10^−5^ M, O_2_ pressure in the system: 1 atm, and sample volume: 4 mL.

Initiation rate *W_i_* was calculated according Equation (3) from a known AIBN concentration in a sample:*W_i_* = *ek_p_*[AIBN](3)

*e:* radicals’ yield relative to one decaying initiator molecule*, k_p_*: an initiator decay rate constant; for cumene at 60 °C: *e =* 1.13*, k_p_* = 1.01 × 10^−5^ s^−1^, *W_i_* = 6.09 × 10^−8^ M·s^−1^.

Absolute values of *k_7_* were calculated taking into account that chain continuation rate constant *k_2_* was 1.75 M^−1^⋅s^−1^ under the model conditions of oxidation in question [55]. 2,6-Di-*tert*-butyl-4-methylphenol (BHT) served as a reference standard.

EPR spectra of phenoxyl–nitroxides **22** were acquired by means of X-band CW EPR spectrometer Bruker Elexsys E540 at 295 K for diluted (10^−4^–10^−5^ M) and oxygen–free-radical solutions. Experimental settings were as follows: microwave power, 0.8 mW; modulation frequency, 100 kHz; modulation amplitude, 0.02 mT; number of points, 1024; and the number of scans, 10. For determining radicals’ decomposition rates, the corresponding EPR spectra were recorded every 10 min for 1–4 h at the following experimental settings: microwave power, 2.0 mW; modulation frequency, 100 kHz; modulation amplitude 0.02 mT; the number of points, 1024; and the number of scans, 10. To determine the value of isotropic g-factors (g_iso_), X-band CW EPR spectra of a mixture of an analyzed radical with Finland trityl [56] were recorded. Then, the known g_iso_ of Finland trityl was used for the spectrum simulation, and the target g_iso_ value was excluded. The simulations of the solution EPR lines were carried out using the software package *Easy Spin* which is available at www.easyspin.org

Geometry optimization and spin density calculation for **22a–o** were performed by means of Gaussian 09 software package at the UB3LYP/6-31G(d) level of theory.

### 3.2. Synthesis

#### 3.2.1. General Procedure for Formylation of 2,6-dialkylphenols **24a–e** Using the Synthesis of 4-hydroxy-3-cyclohexyl-5-methylbenzaldehyde (**23b**) as an Example.

A mixture of 2-cyclohexyl-6-methylphenol [57] (9.51 g, 50 mmol), hexamethylenetetramine (14.1 g, 100 mmol), 50 mL of glacial acetic acid, and 10 mL of water was placed into a 2-necked round-bottomed flask equipped with a thermometer and a Dean–Stark trap. The reaction mixture was stirred at 105–110 °C until 11 mL of water collected in the receiver, then the mixture was refluxed at 118–120 °C for 6 h. After cooling to ambient temperature, the solution was diluted with the equal volume of water, the formed precipitate was filtered off, washed with ice water (2 × 20 mL), air dried, and twice crystallized: first from benzene and then from EtOH.

Pale yellow crystals, isolated yield 10.21 g (94%), m.p. 131–133 °C (EtOH). Elemental analysis: found: C, 77.05; H, 8.53; calcd. for C_14_H_18_O_2_: C, 77.03; H, 8.31%. UV (EtOH), λ_max_ nm, (lg ε): 230 (4.21), 293 (4.17). ^1^H NMR (400 MHz, CDCl_3_), δ, ppm (*J,* Hz): 1.19–1.31 (1H, m, C_6_H_11_); 1.35–1.48 (4H, m, C_6_H_11_); 1.70–1.79 (1H, m, C_6_H_11_); 1.79–1.91 (4H, m, C_6_H_11_); 2.30 (3H, s, CH_3_); 2.76–2.89 (1H, m, C*H*-(CH_2_)_5_); 5.99 (1H, br. s, OH); 7.51 (1H, d, *J* = 1.8, H-6); 7.59 (1H, d, *J* = 1.8, H-2); 9.79 (1H, s, CHO). ^13^C NMR (100 MHz, CDCl_3_), δ, ppm: 16.0 (CH_3_); 26.0, 26.7, 32.9 (3 CH_2_); 37.0 (*C*H-C=C); 123.8 (C-5); 127.1 (C-2); 129.2 (C-1); 130.5 (C-6); 133.6 (C-3); 157.2 (C-4); 191.7 (CHO).

4-Hydroxybenzaldehydes **23a**, **c–e** were obtained in a similar way. Their spectral characteristics and melting points were in accordance with those described in the literature (see Appendix A).

#### 3.2.2. General Synthetic Procedure for 2-(3,5-dialkyl-4-hydroxyphenyl)-4-aryl-1-hydroxy-4,4-dimethyl-2,5-dihydro-1*H*-imidazoles **20a–s**.

Ammonium acetate (2.78 g, 36 mmol) and corresponding 3,5-dialkyl-4-hydroxybenzaldehyde **23** (6.2 mmol) were successively added to a solution of 2-hydroxylamino ketone **25a–c** [58] as hydrochloride or its free base **25d** [59] (6 mmol) in methanol (5 mL) (or in absolute EtOH in the case of the synthesis of **20p–s**). The reaction mixture was diluted with 5 mL of the corresponding alcohol and stirred for 6–12 h until the full conversion of 2-hydroxylamino ketone (TLC control). The resultant suspension was kept for 12 h at 20 °C and 3 h at 0 °C; the formed precipitate was filtered off; washed with cold alcohol (2 × 4 mL), water (10 mL), and again cold alcohol (3 mL); and air dried at rt. In the case of **20r**, a combined alcohol filtrate was evaporated; the residue was mixed with 25 mL of water and kept for 48 h at 0 °C. The formed precipitate was filtered off and air dried at rt, thereby giving an additional amount of **20r**. To obtain **20q** from the reaction mixture, the solution was evaporated, and the residue was mixed with water (40 mL) and incubated for 24 h at 0 °C. The formed precipitate was filtered off, washed with water, and air dried until constant weight. To prepare an analytical sample, the dried precipitate was washed thoroughly with 20 mL of CHCl_3_ to remove traces of starting benzaldehyde and 4*H*-imidazole 3-oxide and dried finally at 80 °C.

#### 3.2.3. General Synthetic Procedure for 5-aryl-2-(3,5-dialkyl-4-hydroxyphenyl)-4,4-dimethyl-4*H*-imidazole 3-oxides **21a–s**

To a suspension of 2,5-dihydro-1*H*-imidazole **20** (5 mmol) in 30 mL of MeOH, a solution of copper(II) acetate hydrate (199 mg, 1 mmol) in 16% aq ammonia (2 mL) was added, and the mixture was stirred with bubbling by a slow stream of air at 20 °C during 1–6 h until the full substrate conversion. The formed yellow precipitate of **21** was filtered off, washed with 5% aq hydrochloric acid and water, and dried at 60–70 °C. An additional amount of 4*H*-imidazole 3-oxide (in the case of **21a**, **c**, **e**, **f**, **h**, **m, q**) was obtained by evaporation of the methanolic filtrate, followed by extraction of the residue with CHCl_3_, washing of the organic layer with 5% aq HCl, drying over MgSO_4_, evaporation, and trituration of the residue with hexane. In the case of isolation of imidazoles **21p**, **r**, **s**, the reaction mixture was evaporated, and the residue was dissolved in 30 mL of CHCl_3_, washed with 5 mL of 5% aq HCl, dried, and evaporated. The residue was purified by column chromatography (SiO_2_, eluent CHCl_3_: MeOH, 20:1), the bright yellow fraction was collected, and after evaporation and trituration of the residue with hexane, a precipitate of 4*H*-imidazole 3-oxide **21p**,**r**,**s** was filtered off.

#### 3.2.4. Preparation of Hybrid Phenoxyl–Nitroxides **22a–o**

Formation of HPNs **22a–o** was accomplished in the following way: PbO_2_ (5 mg) was added to a solution of diamagnetic precursor **21a–o** (~3 mg) in CHCl_3_ (2 mL), and the mixture was stirred for 1 min at 295 K. An inorganic precipitate was filtered off, and 5 μL of the filtrate were taken, diluted with 1 mL of toluene, and bubbled with argon.

## 4. Conclusions

In summary, a large set of cyclic hydroxylamines of the 2,5-dihydroimidazole series was obtained via condensation of *para*-formyl–substituted 2,6-dialkylphenols with different aromatic 2-hydroxylamino ketones containing donor and acceptor substituents on the benzene ring. Their mild oxidation allowed to obtain the corresponding 4*H*-imidazole 3-oxides capable of generating persistent hybrid phenoxyl-nitroxide radicals under the conditions of heterogenic oxidation. It was shown that the antioxidant activity of the investigated imidazole derivatives strongly depends on i) the number of active centers in a molecule able to react with ROS; ii) the structure and effective volume of alkyl substituents in the phenolic moiety, making 1-hydroxy-2-(3,5-dialkyl-4-hydroxyphenyl)-2,5-dihydroimidazole derivatives more active than the widely known BHT. On the other hand, it was established that steric shielding of the phenoxyl moiety influences the hybrid radical stability more strongly than the nature of the substituent at the *para*-position of the aryl substituent at position C-5 of the heterocycle.

Considering that stable phenoxyl radicals and their diamagnetic precursors have a high potential in modern applications of materials chemistry (as electroactive elements [60], additives for the prevention of destruction of perovskite-derived solar cells [61], and effective hydrogen acceptors in ammonia fuel cells, where electricity is generated through oxidation of NH_3_ to dinitrogen [62]), we suppose that hybrid phenoxyl–nitroxides and their precursors can be considered in the future as promising compounds for new technologies of energy storage and processing.

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
