# Peer review of "5-Aryl-2-(3,5-dialkyl-4-hydroxyphenyl)-4,4-dimethyl-4H-imidazole 3-Oxides and Their Redox Species: How Antioxidant Activity of 1-Hydroxy-2,5-dihydro-1H-imidazoles Correlates with the Stability of Hybrid Phenoxyl–Nitroxides"

_molecules, 2020, doi:10.3390/molecules25143118_

Round 1
Reviewer 1 Report
The paper of Mazhukin and co-workers reports a set of cyclic hydroxylamines of the 2,5-dihydroimidazole series and the corresponding 4H-imidazole 3-oxides. The antioxidant activity was investigated and correlated to the molecular structures. The paper is well written and the experimental part, corroborated by computational calculations, appears competently carried out. In my opinion the paper could be of interest for the readers of Molecules and I suggest its publication in the present form.
Author Response
Thank you for your appreciation of our manuscript.
Reviewer 2 Report
This paper presents the preparation of imidazole-based cyclic nitrones and the application of their nitroxyl radical derivatives as antioxidants. The synthesis of these derivatives has been efficiently planned and realized using traditional approaches pertaining to the nitrone chemistry. Evaluation of the antioxidant proprieties of the obtained nitrones allowed establishing some interesting structure-activity relationships which can be useful for potential practical utilizations. Considering the general interest on the chemistry of nitrones my opinion is that this contribution fully deserves publication in Molecules essentially as it stands.
Author Response

(The authors gave the same response as above.)

Reviewer 3 Report
This manuscript describes an interesting, original study and is generally written in a clear and accurate way. I just have some comments, that are reported in the attached list, together with the necessary corrections.

Reviewer 4 Report
In this manuscript Dmitrii G. Mazhukin et all prepare different 1-hydroxy-2-(3,5-dialkyl-4-hydroxy-phenyl)-2,5-dihydroimidazole derivatives and by oxidation the corresponding 4H-imidazole 3-oxides capable of generating persistent hybrid phenoxyl-nitroxide radicals under the conditions of heterogenic oxidation which have a high potential in modern applications of materials chemistry.
In my opinion, the article deserves to be published on molecules.
Minor points:
Pg. 5. Line 146. The yield in the obtention of 21 is missed.
Pg. 6. Line 174. When referring to Scheme S1 in supporting, it must be corrected, since it is confusing because on pg. S1 there is another Scheme S1 and here it refers to the one on pg. S12.
